# Extraction of Abandoned Land in Hilly Areas Based on the Spatio-Temporal Fusion of Multi-Source Remote Sensing Images

**Shan He [1], Huaiyong Shao [1],\*, Wei Xian [2], Shuhui Zhang [1], Jialong Zhong [3] and Jiaguo Qi [4]**

[1]  Key Laboratory of Geoscience Spatial Information Technology, Ministry of Land and Resources of China, Chengdu University of Technology, Chengdu 610059, China; cduthes1991@gmail.com (S.H.); zshcdut0301@gmail.com (S.Z.)

[2]  College of Resources and Environment, Chengdu University of Information Technology, Chengdu 610225, China; xianwei@cuit.edu.cn

[3]  College of Management Science, Chengdu University of Technology, Chengdu 610059, China; jialogn@gmail.com

[4]  Center for Global Change and Earth Observations, Michigan State University, East Lansing, MI 48824, USA; qi@msu.edu

\*  Correspondence: shaohuaiyong@cdut.edu.cn

**Abstract:** Hilly areas are important parts of the world's landscape. A marginal phenomenon can be observed in some hilly areas, leading to serious land abandonment. Extracting the spatio-temporal distribution of abandoned land in such hilly areas can protect food security, improve people's livelihoods, and serve as a tool for a rational land plan. However, mapping the distribution of abandoned land using a single type of remote sensing image is still challenging and problematic due to the fragmentation of such hilly areas and severe cloud pollution. In this study, a new approach by integrating Linear stretch (Ls), Maximum Value Composite (MVC), and Flexible Spatiotemporal DAta Fusion (FSDAF) was proposed to analyze the time-series changes and extract the spatial distribution of abandoned land. MOD09GA, MOD13Q1, and Sentinel-2 were selected as the basis of remote sensing images to fuse a monthly 10 m spatio-temporal data set. Three pieces of vegetation indices (VIs: ndvi, savi, ndwi) were utilized as the measures to identify the abandoned land. A multiple spatio-temporal scales sample database was established, and the Support Vector Machine (SVM) was used to extract abandoned land from cultivated land and woodland. The best extraction result with an overall accuracy of 88.1% was achieved by integrating Ls, MVC, and FSDAF, with the assistance of an SVM classifier. The fused VIs image set transcended the single source method (Sentinel-2) with greater accuracy by a margin of 10.8–23.6% for abandoned land extraction. On the other hand, VIs appeared to contribute positively to extract abandoned land from cultivated land and woodland. This study not only provides technical guidance for the quick acquirement of abandoned land distribution in hilly areas, but it also provides strong data support for the connection of targeted poverty alleviation to rural revitalization.

**Keywords:** abandoned land; cloud pollution; hilly area; multi-source images; spatio-temporal fusion; time-series change

## 1. Introduction

The hilly area is the transition zone between the mountain and the plain. Sloping land is the most prevalent land type, while some sloping areas show the characteristics of fragmented land, a staggered distribution of land cover types, diverse crop types, and complex planting structures [1]. Some hilly areas are covered by clouds for a long time. These complicated circumstances have caused topographical marginality and serious abandonment of land [2]. The abandoned land in hilly areas has a significant negative impact on countries and regions with limited per capita cultivated land resources and

a large proportion of sloping cultivated land (such as China) [1,3,4]. The development of effective methods to extract the distribution of abandoned land in hilly areas and to quickly produce a temporal and spatial distribution map is of great significance to protect food security on a local and global scale, to rationally plan land, and to improve people's livelihoods [5,6].

Remote sensing has been widely used to identify abandoned land around the world (Table 1). On the one hand, land abandoned in a larger region is most commonly mapped using low-resolution satellite images such as those produced by MODIS, which provides consecutive images with a high temporal resolution. These studies are of great significance for large-scale farmland abandonment research, but for local farmland abandonment, or fragment abandoned land, higher-precision remote sensing images with a spatial and temporal resolution are required.

On the other hand, abandoned land in a small area is usually mapped using satellite images with a medium resolution. Based on this idea, some scholars have used Landsat/HJ-1A remote sensing data to extract a 30 m resolution abandoned land distribution map. These studies have made a remarkable contribution to the study of local abandoned land. However, regardless of the 250 m–1 km resolution of the MODIS image or the 30 m resolution of the Landsat/HJ-1 image, their pixel size was larger than the size of abandoned land in some hilly areas, which brought errors in extracting fragmented abandoned land. Furthermore, cloud pollution was an significant factor in their research, resulting in a reduction in the available images and a decrease in accuracy. For such hilly areas, a higher spatio-temporal resolution of remote sensing images is needed to increase the available images and to improve the extraction accuracy of abandoned land. The Sentinel-2A satellite was launched in 2015 and the Sentinel-2B satellite was launched in 2017. Their highest spatial resolution is 10 m, and the revisit period is 5 days. The Sentinel-2 image has been widely used in land use and coverage changes. However, few studies have extracted abandoned land based on Sentinel-2 images. Heeyeun Yoon extracted abandoned land from dry land and paddy fields in Gwangyang City, South Korea from 2016 to 2018 by using three pieces of vegetation indices (VIs: ndvi, ndwi, and savi) from Sentinel-2 images and SVM classification [7]. His team used the harmonic function of VIs to improve the accuracy of abandoned land extraction, with an accuracy of 90.72%. Although the improvement in the accuracy has been verified using the three pieces of vegetation indices combined with SVM classification, the descending number of available remote sensing images and cloud interference have remained as obstacles for data quality.

In order to obtain a more accurate temporal and spatial distribution of abandoned land, especially in hilly areas with fragmented land and severe cloud interference, combining the temporal resolution of MODIS and the spatial resolution of Sentinel-2 appears to be a viable and reasonable option. The spatial and temporal adaptive reflectance fusion model (STARFM) blends Landsat and MODIS data to predict daily surface reflectance at the Landsat spatial resolution and the MODIS temporal frequency [8]. STARFM has been widely used in land surface coverage and change monitoring and has achieved a series of good application effects [9–12]. Subsequently, many algorithms based on STARFM improvement or space-time fusion based on MODIS and Landsat have been proposed, and a certain degree of improvement has been achieved in different usage scenarios [13–20]. Among them, the Flexible Spatiotemporal DAta Fusion (FSDAF) demonstrated promising advantages in extracting vegetation changes in fragmented heterogeneous regions, and it can obtain relatively good results depending on algorithm stability, data fusion accuracy, and fusion efficiency [21–23]. Based on FSDAF, Liu developed the Improved Flexible Spatiotemporal DAta Fusion (IFSDAF) method using all available finer-scaled images, including those partly contaminated by clouds, to improve the accuracy [24]. Although the IFSDAF algorithm makes maximum use of the pixels of high spatial resolution images, it fails to analyze the changes of images in different periods. As the radiation difference of images in different periods is large, cloud-free pixel filling may form clumpy errors, which may distort the classification results.

In this study, abandoned land was defined as cultivated land without cultivation or woodland damaged due to felling, fire, and other factors for more than one year. Linear stretching was used to smooth the difference in radiation between the different images, and the MVC was used to obtain the largest monthly planting area in the study area. A new approach of integrating Linear stretch (Ls), Maximum Value Composite (MVC), and Flexible Spatiotemporal DAta Fusion (FSDAF) was proposed to analyze the time-series changes and to extract the spatial distribution of the abandoned land. It retains the pixels of the Sentinel-2 image to the greatest extent and obtains the surface changes of the MODIS images under the Sentinel-2 cloud area. In order to avoid error transmission during fusion, we first extracted the vegetation indices (VIs) and then used the integrated Ls, MVC, and FSDAF (Ls+MVC+FSDAF) to obtain the monthly VIs data set [25]. Combining the land use distribution map in 2018, the land cover product at 30 m in 2020 [26–30], Openstreetmap vector, Google orthophoto, a UAV image, and time-series images of Sentinel-2, MOD09GA, and MOD13Q1, the Support Vector Machine (SVM) was used to extract the abandoned land distribution. The goal of providing data support and effective suggestions to the local land management department can be achieved by extracting the distribution of abandoned land.

**Table 1.** Applied remote sensing data to extract the abandoned land.

| Used Remote Sensing Data | Number of Studies | Study IDs |
| --- | --- | --- |
| MODIS | 9 | [4,31–38] |
| Landsat | 17 | [3,39–54] |
| Sentinel-2 | 1 | [7] |
| SPOT | 2 | [55,56] |
| RapidEye | 1 | [57] |

## 2. Materials and Methods

The technical flowchart of the study includes three steps (Figure 1). The main purpose of step 1 was to extract the range of woodland and cultivated land, and samples of woodland, cultivated land, and abandoned land in the ArcGIS platform, including the following aspects: (i) Using attribute information to screen woodland and cultivated land on the land use distribution map and land cover product, and then the results were overlay analyzed to improve the accuracy; (ii) the spatial resolution of the above data was 30 m, so it was necessary to screen the woodland and cultivated land range. Google orthophoto has a spatial resolution of 1.7 m and has a good recognition ability for small plots. In the range of obtained woodland and cultivated land, human–computer interaction was carried out on Google orthophoto to eliminate the range that did not belong to woodland and cultivated land. The OpenStreetMap vector can provide the distribution of roads, buildings, water, and riverbanks, etc., which was used to mask the overlapped result. Overlying the Google orthophoto human–computer interaction results and the Openstreetmap vector eliminated the range that did not belong to woodland and cultivated land. (iii) The screened distributions of woodland and cultivated land were overlayed on the true color image of sentinel-2 to find areas that may be abandoned. A total of 9 typical areas and planned field sampling routes were delineated. Finally, the samples of forest land, cultivated land, and abandoned land were obtained by time-series Sentinel-2 image delineation, Google orthophoto human–computer interaction, UAV operation in the 9 typical areas, and field sampling in the study areas.

The main purpose of step 2 was to integrate Ls, MVC, and FSDAF to produce the monthly cloud-free 10 m spatial resolution of the VIs image set, including the following aspects: (i) extracting the VIs image set from Sentinel-2, MOD13Q1, and MOD09GA separately; (ii) performing Ls, MVC, and FSDAF to obtain a 10 m resolution VIs image set (iii) combining Ls+MVC+FSDAF VIs image set and Sentinel-2 cloud-free VIs to obtain a monthly 10 m resolution VIs image set.

The main purpose of step 3 was to analyze the VIs time-series of the woodland, cultivated land, and abandoned land and extract the distribution of the abandoned land in the study areas.

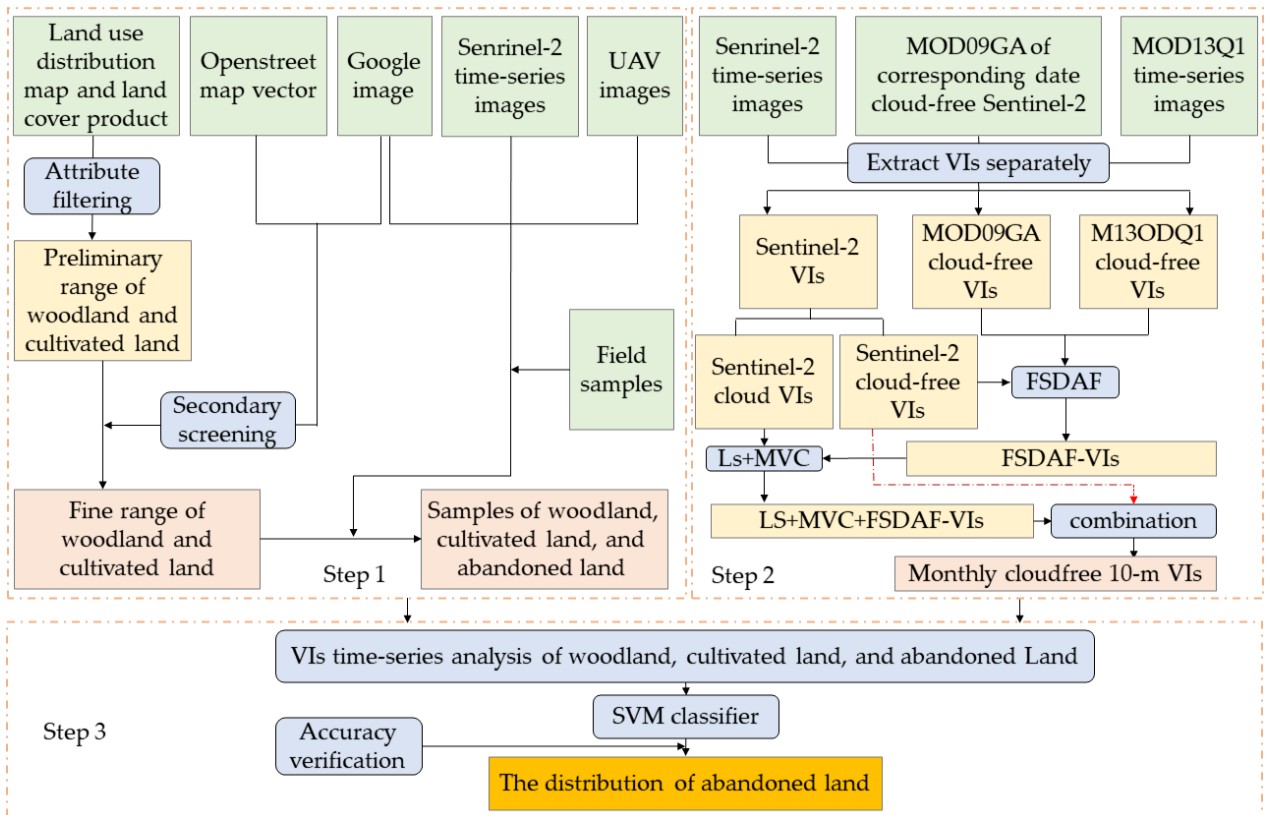

**Figure 1.** Technical flowchart.

### 2.1. Study Area

The study area (105.056111 N~105.489121 N, 30.429114 E~30.733959 E) is located in the middle of the Sichuan Basin, the middle reaches of the Fujiang River, and belongs to Daying county, Suining city. It borders Pengxi county in the east, Chuanshan district and Anju district in Suining city in the south, Lezhi county in Ziyang city, Zhongjiang county in Deyang city in the west, and Shehong city and Santai county in Mianyang city in the north (Figure 2). Daying county is dominated by hilly areas, with a total area of 701 km$^2$. The relatively fragmented land and the low level of mechanization facilitated the occurrence of the abandonment phenomenon.

The crops in the Daying county are mainly planted in the two phenological seasons of "spring and late autumn", during which rice, corn, peanut, soybean, and sweet potato are planted in the spring (from April to September), and winter wheat, rape, and potato are planted in the late autumn (from October of the first year to March of the second year). The crop phenological cycle is shown in Figure 3.

### 2.2. Data and Preprocessing

#### 2.2.1. Data Source

The 19 L2A-level Sentinel-2 images with cloud cover of less than 80% were downloaded from the official website of ESA [58], and 8 MOD09GA images and 23 MOD13Q1 images were downloaded from the official website of USGU [59], the land cover product in 2020 of the study area was downloaded the from Earth Science Big Data Science Engineering Data Sharing Service System [60], the Openstreetmap vector of the study area was downloaded from the Openstreetmap official website [61], and Google orthophoto was

downloaded from the Bigmap platform. The land use distribution map in 2018 was obtained from the Sichuan Provincial Department of Natural Resources. The remote sensing data and its parameters are shown in Table 2.

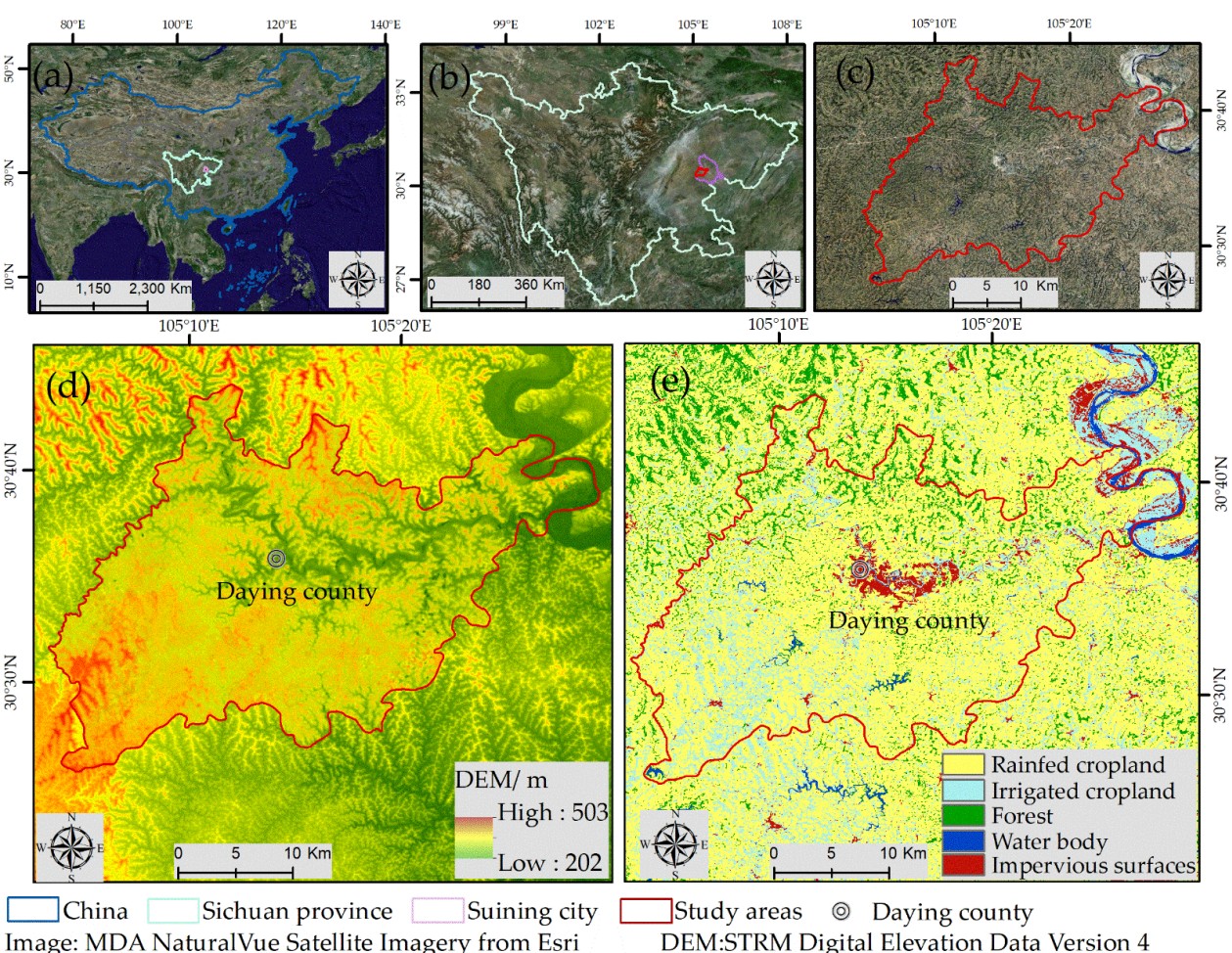

**Figure 2.** (**a**) The location of the study areas in China, (**b**) The location of the study areas in Sichuan province, (**c**) the enlarged map showing the study areas, (**d**) Digital Elevation Model (DEM), and (**e**) land cover product in the study areas.

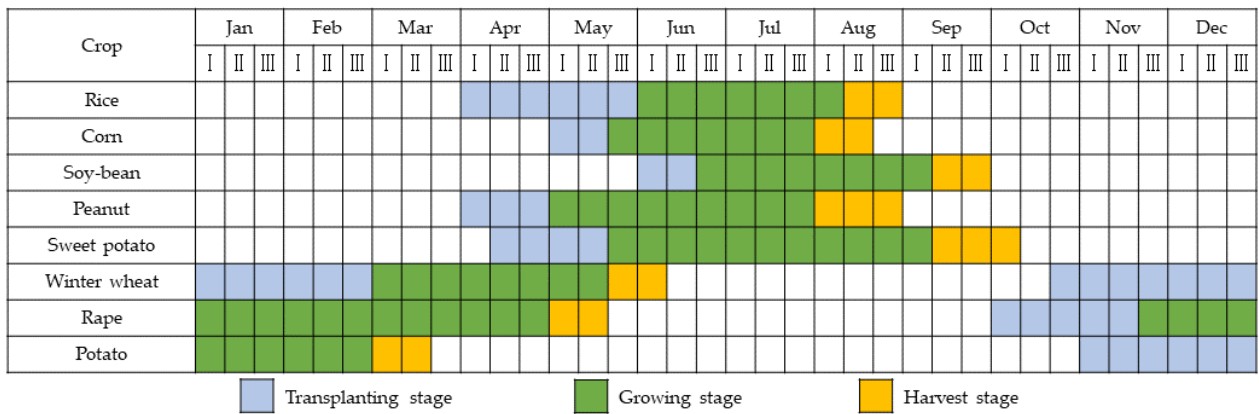

**Figure 3.** The phenological cycle of the main crops in the study area.

<div align="center">**Table 2.** Remote sensing data and parameters.</div>

| Remote Sensing Type | Band Number | Band Range (nm) | Spatio-Temporal Resolution (d/m) |
|---|---|---|---|
| Sentinel-2 | Band 4—Red | 650–680 | 5/10 |
|  | Band 8—NIR | 785–900 | 5/10 |
|  | Band 12—SWIR | 2100–2280 | 5/20 |
| MOD09GA | Band 1—Red | 620–670 | 1/250 |
|  | Band 2—NIR | 841–876 | 1/250 |
|  | Band 7—SWIR | 2105–2155 | 1/500 |
| MOD13Q1 | ndvi | - | 16/250 |
|  | Band 1—Red | 620–670 | 16/250 |
|  | Band 2—NIR | 841–876 | 16/250 |
|  | Band 7—SWIR | 2105–2155 | 16/500 |

Since the accuracy of the samples directly affects the extraction results of the abandoned land, it was necessary to collect the samples at multiple temporal and spatial scales. Cultivated land planting shows obvious periodicity, which can be identified by the planting cycle of crops in the time-series Sentinel-2 true color images. In addition, the color of the cultivated land was lighter, the brightness was higher, the texture continuity was good, and the overall shape was quadrilateral with obvious boundaries. When the image of Sentinel-2 was not clear, the cultivated land could be further identified by Google orthophoto and UAV image (Figure 4(c1,c2)). The color of woodland was dark green in the Sentinel-2 true-color image from June to August, and it was generally located at a relatively high altitude. Meanwhile, woodland had a certain cluster shape on the Google orthophoto and the UAV image, which could be further identified (Figure 4(b1,b2)). The abandoned land was light green to dark brown, with mottled texture features and no obvious boundary lines. It is easy to be confused with cultivated land and woodland in recognition. For further identification, the time-series Sentinel-2 true color images were used for preliminary screening, and the Google orthophoto and UAV image were also used (Figure 4(b1–b3,c1–c3)).

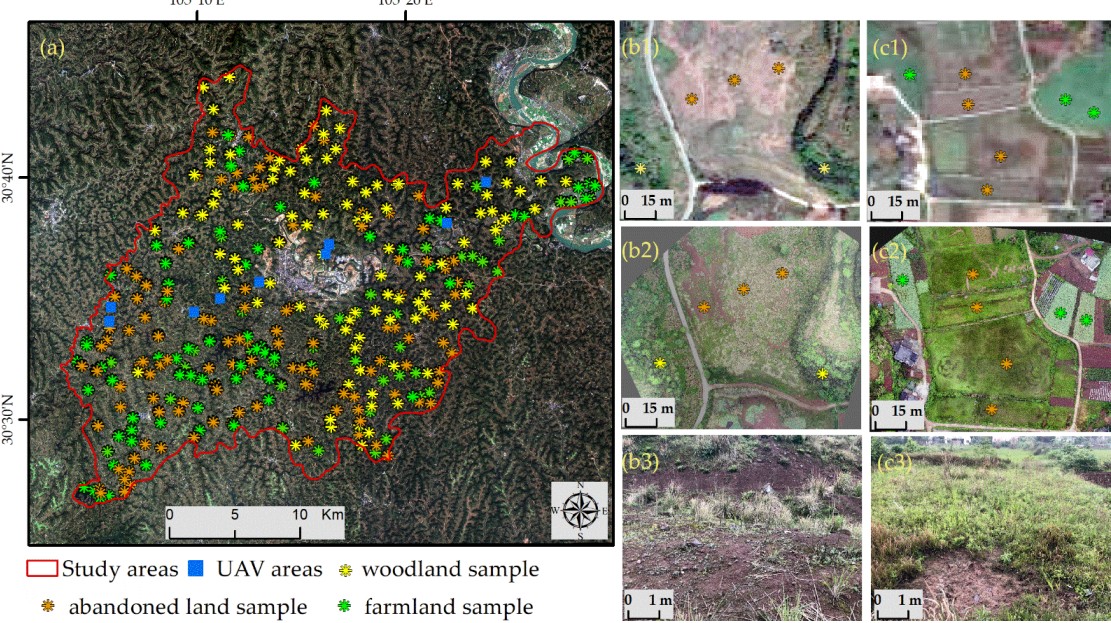

**Figure 4.** (**a**) is the spatial distribution of the sample points in the Sentinel-2 image. (**b1–b3**) is the appearance of the abandoned land sample in the woodland in Google orthophoto, the UAV images, and the ground photography, respectively. (**c1–c3**) is the appearance of the abandoned land sample in the farmland in Google orthophoto, the UAV images, and the ground photography, respectively.

Firstly, the Sentinel-2 true-color images from January to December and the Google orthophoto were employed to select a total of 300 samples, including 100 woodland samples, 100 cultivated land samples, and 100 abandoned land samples. Secondly, based on the image characteristics and preliminary data collection, Phantom 4 pro was selected to conduct drone operations in 9 typical areas, with a flying height of 300 m. Among the vector sketching samples, a total of 34 samples of woodland, 36 samples of cultivated land, and 27 samples of abandoned land were obtained. Meanwhile, the GPS handsets were used to sample 11 villages and towns across the study area. A total of 48 samples of woodland, 67 samples of cultivated land, and 39 samples of abandoned land were obtained.

The obtained samples were screened with a large space between the samples and discretely distributed at various locations in the study area to reduce the correlation between the samples. Subsequently, an extension of the Jeffries–Matusita distance was used to measure class separability among the screened samples [62]. After calculation, the separability between the three kinds of samples was greater than 1.9, indicating that the samples after screening were well separable. The sample database was generated through the combination of the time-series Sentinel-2 true-color image, the Google orthophoto, the UAV image, and the ground sampling, which increases the diversity and reliability of samples (Figure 4).

### 2.2.2. Data Processing

Sentinel-2 images were L2A level, MOD09GA images were L2G level, and MOD13Q1 images were L3 level. They all had undergone atmospheric correction and geometric precision correction. The downloaded images were then processed as follows:

(1) Projection transformation: MOD09GA and MOD13Q1 images were uniformly transformed into the same projection coordinate system as Sentinel-2 images (WGS 84/UTM zone 48);

(2) Resample: MOD09GA and MOD13Q1 images in the near-infrared band and red band were resampled to a spatial resolution of 10 m, and the mid-infrared band to 20 m. The method of resample was a bilinear interpolation;

(3) Vector crop: The MOD09GA and MOD13Q1 images were cropped by Sentinel-2, and all images were the same size;

(4) Image registration: The Sentinel-2 image was used as a reference to correct the MOD09GA and MOD13Q1 images;

(5) Band calculation: The ndvi, savi, and ndwi of Sentinel-2 and MOD09GA and MOD13Q1 images were calculated, respectively:

$$ndvi = \rho_{(N)} - \rho_{(R)} / \rho_{(N)} + \rho_{(R)} \tag{1}$$

$$ndvi = \rho_{(N)} - \rho_{(M)} / \rho_{(N)} + \rho_{(M)} \tag{2}$$

$$savi = (\rho_{(N)} - \rho_{(R)}) * (1 + L) / (\rho_{(N)} + \rho_{(R)} + L) \tag{3}$$

where, $\rho_{(M)}$ is the mid-infrared band, corresponding to band 7 of the MODIS image and band 12 of the Sentinel-2 image; $\rho_{(N)}$ is the near-infrared band, corresponding to band 2 of the MODIS image and band 8 of the Sentinel-2 image; $\rho_{(R)}$ is the red band, corresponding to band 1 of the MODIS image and band 4 of the Sentinel-2 image, respectively (Table 2); L is the soil adjustment coefficient, with a value of 0.5.

Spatio-temporal fusion: Ls, MVC, and FSDAF fusion was performed on ndvi and savi (shown in 2.3) to obtain a 10 m spatial resolution image, and Ls, MVC, and FSDAF fusion were performed on ndwi to obtain a 20 m spatial resolution image and were resampled to 10 m spatial resolution. The method of resample was a bilinear interpolation.

Mask: The other class type was masked apart from the woodland and cultivated land.

### 2.3. Data Combining

#### 2.3.1. Ls+MVC

The maximum value composite (MVC) utilizes cloud detection and quality inspection on the remote sensing image to compare the vegetation index image value pixel by pixel and selects the maximum value as the synthesized result [63]. It merges the vegetation index of the remote sensing images at different times to obtain the maximum vegetation coverage during that period [64]. At the same time, the technique has also been applied with the Moderate Resolution Imaging Spectroradiometer (MODIS) instrument, in combination with other constraints to help exclude other undesirable artefacts from extreme view angles, cloud contamination, and other sources, producing 16-day vegetation index composites [65]. The monthly composite VIs images show to the greatest extent the largest area of crops planted in the remote sensing image that month. However, the various geometric positions of the sun corresponding to the remote sensing images at different times, and the change in the amount of radiation from the surface cover over time led to differences in the amount of radiation from different remote sensing images. If the difference is not eliminated but the MVC is performed directly, a clumpy error may be formed (Figure 5 MVC 1/2/3).

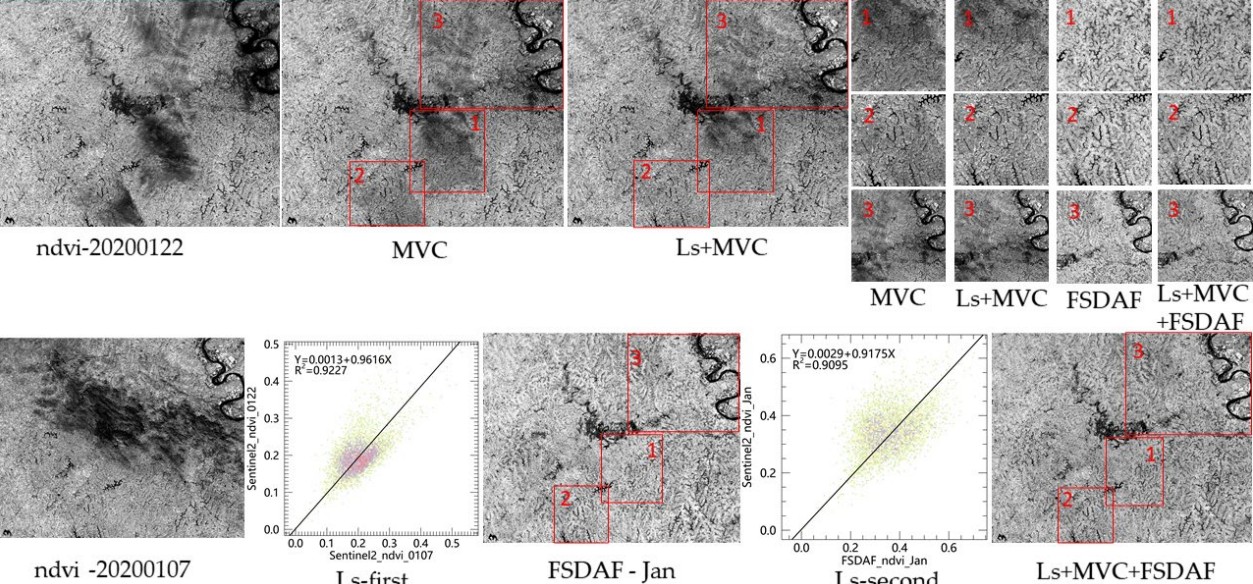

**Figure 5.** The integration of Ls and MVC to combine the images.

Two remote sensing images of the same sensor at different times in the same area or different sensors at the same time in the same area were compared. A correlation analysis can usually be used to obtain the relationship between the two, including linear models, quadratic models, exponential models, geometric models, hyperbolic models, and the logarithmic square model. Linear stretch is a common method to compare the differences between two images. The radiation difference between the different images can be eliminated by using the linear regression model [16], and the smooth transition of the same ground object and the variances of different ground objects can be achieved [66]. Therefore, linear stretch was used between the remote sensing images in the above two situations to obtain the correlation and the coefficient of determination $R^2$ was evaluated after the correlation.

$$Y = a * X + b \tag{4}$$

where $Y$ is the reference image, $X$ is the image to be stretched, and $a$ and $b$ are the correlation coefficients.

$$R^2 = 1 - \sum_{i=1}^{n} (y_i - \hat{y}_j)^2 / \sum_{i=1}^{n} (y_i - \bar{y})^2 \tag{5}$$

where $y_i$ is the pixel value of the reference image, $\hat{y}_j$ is the pixel value of the image to be stretched, $\overline{y}$ is the average pixel value of the reference image, and n is the number of pixels in the study area. The value range of $R^2$ is 0–1. The larger the value of $R^2$, the higher the correlation between the two images.

Combining MVC and Ls can eliminate the clumpy error of MVC. The band was combined and ndvi was taken as an example in this study (2.3.2 and 2.3.3 are the same). The red box in Figure 3 shows the cloud coverage area. By observing and comparing the performance of the image in the red boxes after combination, the effect of the combined can be evaluated. The scene classification map of the Sentinel-2 image was used to merge the cloud areas and then the merged cloud areas were used to mask the images to obtain cloud-free pixel images of the same size. The image with a relatively little amount of cloud (Figure 5 ndvi_20200122) was taken as a reference image (y-axis), and the other image (Figure 5 ndvi_20200107) was taken as the image to be stretched (x-axis) to obtain the linear regression equation (Figure 5 Ls-first). The stretched image (Figure 3 ndvi_20200107) was obtained as the linear regression equation was applied. Finally, the stretched image and the reference image were used for MVC to obtain the combined image (Figure 3 Ls+MVC). By comparing the changes in the cloud coverage area between the MVC image and the Ls+MVC image (Figure 3 MVC 1/2/3 and Ls+MVC 1/2/3), it was evident that the Ls+MVC image had been removed from the clumpy shadow in the MVC image and a good result was obtained.

### 2.3.2. FSDAF

FSDAF is a multi-source remote sensing spatio-temporal fusion algorithm that combines unmixing, spatial interpolation, and similar neighboring pixel smoothing to obtain robust fusion results. It can be used to obtain land surface information of gradual changes or sudden changes in land cover types in heterogeneous regions [20]. Firstly, FSDAF estimates the temporal variation of Sentinel-2 pixels ($\triangle F^{tp}$) based on the unmixing of the entire image to generate the temporal prediction ($F_2^{tp}$). Secondly, using the thin plate spline interpolation to generate spatial prediction ($F_2^{SP}$), the residuals between the Sentinel-2 pixels and the MODIS pixels are considered in FSDAF as [23]:

$$R(x,y) = \Delta C(x,y) - \frac{1}{n}\left[\sum_{i=1}^{n} F_2^{tp}(x_i,y_i) - \sum_{i=1}^{n} F_1(x_i,y_i)\right] \tag{6}$$

where $R(x,y)$ is residual in the MODIS pixel at the location $(x,y)$, $n$ is the number of Sentinel-2 pixels inside a MODIS pixel, and the Sentinel-2 pixel at location $(x_i,y_i)$ is inside the MODIS pixel at location $(x,y)$. In a homogenous area, the spatial prediction performs well, which is applied to calculate a new residual [23]:

$$R_{ho}(x,y) = F_2^{sp}(x,y) - F_2^{tp}(x,y) \tag{7}$$

A weighted function ($w_h$) is used for a homogeneity index of residual compensation to integrate the two residuals (i.e., $R_{ho}$ and $R$). The final prediction of FSDAF can be expressed as [23]:

$$\hat{F}_2(x,y) = F_1(x,y) + \sum_{i=1}^{n_s} W_i\big(\Delta F^{tp}(x_i,y_i) + n \times R(x_i,y_i) \times w_h(x_i,y_i)\big) \tag{8}$$

where $W_i$ is the weight of similar pixels, and $\hat{F}_2(x,y)$ is the predicted image.

As shown in Figure 5, the ndvi of the Sentinel-2 March image and the corresponding ndvi of the MOD09GA image were used as the reference image pair, the ndvi of MOD13Q1 January was used as the predicted time-image, and FSDAF was used to obtain the fused image (Figure 5 FSDAF–Jan). The FSDAF smoothed the neighboring pixels, which made the surface coverage information smooth as well. Some errors were caused during FSDAF by various factors such as the radiation difference between Sentinel-2 and MODIS,

spectral difference, spatial resolution ratio, geometric registration error, etc. [23] (Figure 5: FSDAF1/2/3 and Ls+MVC+FSDAF1/2/3). In order to reduce the errors caused by time changes, the time of the reference image and the predicted image should be as semblable as possible [67]. The reference images were selected as follows: January to April was based on March, May to June was based on June, July to August was based on August, and September to December was based on November. Among them, August and November were cloud-free images, and March and June were obtained by Ls+MVC.

### 2.3.3. Ls+MVC+FSDAF

Ls+MVC can make full use of the cloud-free pixels in Sentinel-2 images, but when the clouds are present in the same areas in both images, cloud pollution will become a critical issue. FSDAF can use MODIS cloud-free images, but when the changing area is small, the predicted image may not be able to capture the changes. Ls+MVC+FSDAF can make full use of high spatial resolution image pixels, and obtain fused images of cloud changes, thereby obtaining a high-precision fusion image (Figure 5 Ls+MVC+FSDAF). As shown in Figure 6, we obtained a monthly ndvi of the Sentinel-2 images through Ls+MVC, including cloudy images (Figure 6 Sentinel2-cloud-ndvi) and cloud-free images (Figure 6 Sentinel2-cloudfree-ndvi). Based on the reference image pair composed of Sentinel2-cloudfree-ndvi and the corresponding ndvi of MOD09GA, the monthly ndvi synthesized by MOD13Q1 was used as the predicted image, and the fused image was obtained by using FSDAF (Figure 6 FSDAF_ndvi). Taking the Sentinel-2 image as the reference, a linear regression analysis was performed on the fused image, and the linear regression equation was obtained. The fused image was stretched by the linear stretching equation, and the stretched pixels were filled into the cloud area of Sentinel-2 to obtain a complete cloud-free image. We combined the cloud-free images of Sentinel-2, Ls+MVC images, and Ls+MVC+FSDAF images to obtain a monthly cloud-free VIs image set (Figure 6 Ls+MVC+FSDAF).

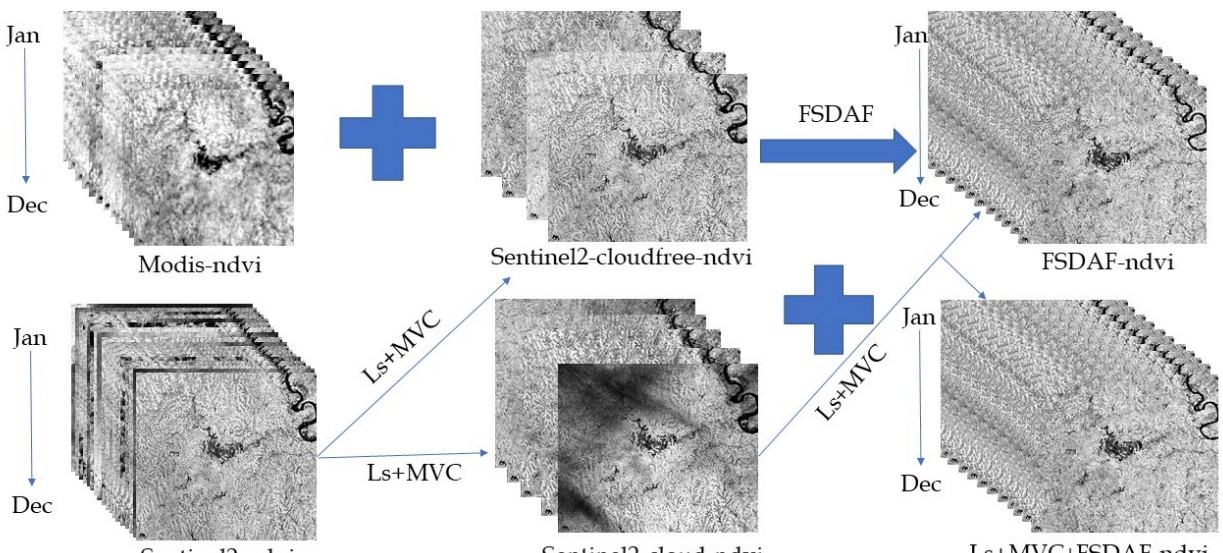

**Figure 6.** Spatio-temporal fusion with Ls+MVC+FSDAF multi-source images.

Through remote sensing image processing, the images were divided into three categories to test whether the fused images could improve the extracted accuracy of the abandoned land: (i) the cloud-free Sentinel-2 images set without image fusion (August and November); (ii) cloud-free image set with Ls+MVC (March, June, August, November); (iii) cloud-free image set with Ls+MVC+FSDAF (January to December).

### 2.4. SVM Classification

SVM is a machine learning algorithm based on statistical learning theory, VC dimension theory, and the structural risk minimization principle. It is often used to solve few samples, nonlinear problems, and high-dimensional pattern recognition problems. SVM uses edge samples between different categories to find the optimal hyperplane between different categories for division. On the premise of the limited classification samples, it can well balance the complexity and learning ability of the model, and greatly avoid the problems of "over-learning" and "dimension disaster". It ensures that the extreme value solution obtained is the global optimal solution and has a good generalization ability. For the land classification problem, SVM uses the kernel function and penalty variables to transform the low-dimensional linear inseparability problem into the high-dimensional linear separable problem. The kernel function parameter ($\gamma$) and the penalty factor (C) are set to solve the problem of individual outlier category attribution. In this way, it reaches the goal of automatic recognition of ground object classification. Predecessors have extracted abandoned land based on SVM and obtained good results [1,5,14,29]. In this study, trained samples and verified samples were randomly divided into 7:3. SVM classifiers were selected to identify woodland, cultivated land, and abandoned land in the study area, and the temporal and spatial distribution of abandoned land was extracted.

### 2.5. Accuracy Verification

This study used the verification samples to calculate the confusion matrix after classification. The overall accuracy, user accuracy, product accuracy, and the Kappa coefficient were calculated through the confusion matrix to evaluate the classification results.

The cloud-free Sentinel-2 images set without image fusion (August and November), the cloud-free image set with Ls+MVC (March, June, August, November), and the cloud-free image set with Ls+MVC+FSDAF (1 Month to December) were used to extract the abandoned land. By comparing the classification accuracy and the Kappa coefficient, the role of the VIs image set generated by integrating Ls, MVC, and FSDAF in the extraction of abandoned land was assessed.

## 3. Results

### 3.1. VIs Time-Series Curve Analysis

Through the selected samples (Figure 4), we mapped the VIs time-series curve diagrams of woodland, cultivated land, and abandoned land according to the average value of the samples. Their maximum and minimum values with horizontal lines were recorded (Figure 7). According to Figure 7, it can be seen that the VIs values were closely related to the growth cycle of vegetation. April and August are the months when crops grow most vigorously. The time-series changes of the cultivated land VIs showed a "double peak" shape. In comparison, however, the "double peak" shape of ndvi and savi was more obvious with the highest peak being in August, and ndwi was relatively gentle with the highest peak in April, which coincided with the rice irrigation period. The time-series curve shape of ndvi and savi of the woodland was "high in the middle, low on both sides", and the highest value was in August. The ndvi and savi time-series curve shape of the abandoned land was "rising first, then falling", which was similar to the woodland, but its average value was generally lower than the woodland. It is notable that the VIs values of woodland had a small variation range (i.e., small variance), generally fluctuating within 0.15. The abandoned land and the cultivated land had a relatively large range of changes, due to the various planting types of the cultivated land and the complex types of abandoned land in the study area. The seasonal changes of the cultivated land were obvious, and part of the cultivated land was single-crop in spring or late autumn. For example, rice was only cultivated in April, harvested from August to September, and laid idle in other months. Correspondingly, winter wheat was cultivated in November, harvested in May, and idled in other months. The seasonal change of single-crop was apparent, especially before and after the crops were harvested, and the value of the VIs dropped rapidly, which

led to great fluctuations in the cultivated land curve. The vegetation coverage type, and coverage degree of abandoned land was different, and its time-series curve changed greatly. Figure 4 shows two different types of abandoned land (one was abandoned in woodland and the other was abandoned in cultivated land). A high VIs value was observed as the vegetation was lush, and vice versa. The three pieces of Vis (ndvi, savi, and ndwi) were used to monitor the phenological changes of woodland, cultivated land, and abandoned land from three aspects: vegetation growth trend, vegetation growth status under different soil backgrounds, and vegetation water content, so as to extract the abandoned land more effectively from the cultivated land and woodland information.

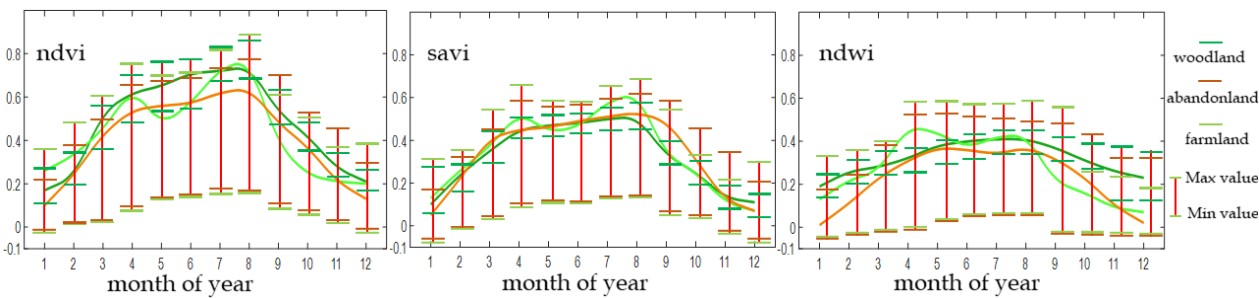

**Figure 7.** VIs timing change curve of different land cover in the study area.

### 3.2. Comparison of Classification Accuracy

The Sentinel-2 image set without data fusion (August, November), the image set with Ls+MVC (March, June, August, November), and the image set with Ls+MVC+FSDAF (January to December) were used to extract the abandoned land in the study area, respectively. The improvement in the extraction accuracy by the integration of Ls, MVC, and FSDAF was evaluated. It can be seen that when the Sentinel-2 image set (August and November) was used solely to extract the abandoned land, it lacked images of the growth period of the key season of the cultivated land, and it was difficult to distinguish the abandoned land from the cultivated land. The image set with Ls+MVC (March, June, August, November) contained the four seasons of spring, summer, autumn, and winter, and could extract the abandoned land from the woodland and cultivated land. However, its temporal resolution was low, only one image was contained in each season, and the extraction accuracy needed to be improved. The image set with Ls+MVC+FSDAF (January to December) increased the available images, generating a VIs image set with a monthly 10 m scale in the study area, which significantly improved the accuracy of the abandoned land extraction (Table 3).

**Table 3.** Accuracy of classification from the different image sets.

| Remote Sensing Image Data Source | Overall Accuracy | User Accuracy | Product Accuracy | Kappa Coefficient |
|---|---|---|---|---|
| Sentinel-2 (August, November) | 64.5% | 54.5% | 57.6% | 0.58 |
| Ls+MVC (March, June, August, November) | 77.3% | 72.7% | 81.4% | 0.73 |
| Ls+MVC+FSDAF (January to December) | 88.1% | 94.1% | 86.5% | 0.87 |

### 3.3. Abandoned Land Distribution

With SVM for classification and the ArcGIS platform for editing, the image set with Ls+MVC+FSDAF was selected to generate the abandoned land distribution map of the study area (Figure 8).

After grid calculation, the abandoned land area in the study area was 6192 ha, accounting for 9.65% of the cultivated land and woodland area.

From Figure 8, it is notable that the distribution of abandoned land in the study area was scattered, and each township had a certain degree of abandonment. In general, the abandonment phenomenon of land is widespread. Among them, Yufeng town had the

highest proportion of abandoned land and Huima town had the lowest. The townships in descending order were Yufeng (12.97%), Xiangshan (11.95%), Zhishui (11.23%), Hebian (10.71%), and Tianbao (10.65%), Jinyuan (10.47%), Zhuotongjing (9.9%), Tongxian (8.97%), Penglai (8.79%), Longsheng (7.7%), and Huima (4.25%). In terms of the abandoned land area, Penglai town had the highest abandoned land area, whereas Zhishui town had the lowest. The townships in descending order were Penglai (997 ha), Hebian (965 ha), Yufeng (810 ha), and Longsheng (738 ha), Xiangshan (602 ha), Jinyuan (534 ha), Tianbao (523 ha), Zhuotongjing (393 ha), Tongxian (269 ha), Huima (187 ha), and Zhishui (174 ha) (Figure 9).

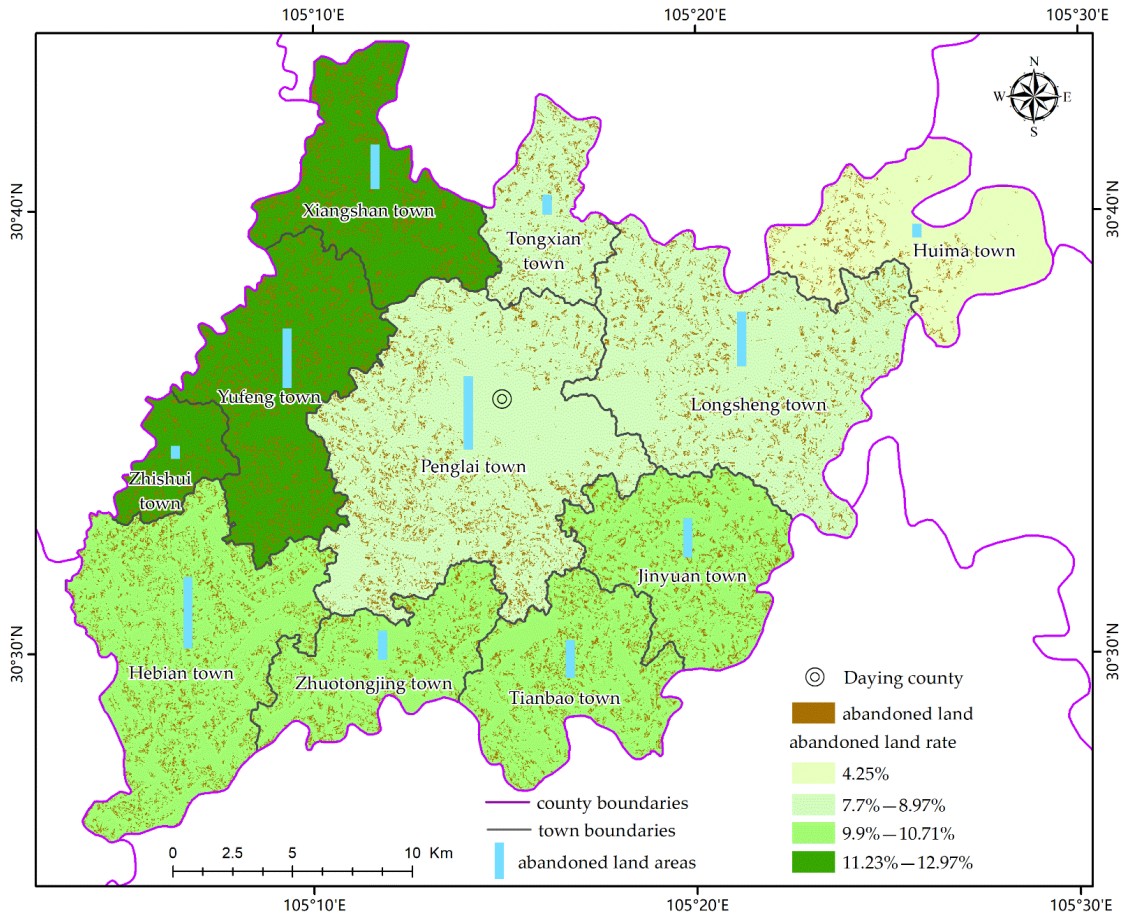

**Figure 8.** Abandoned land distribution map in the study area.

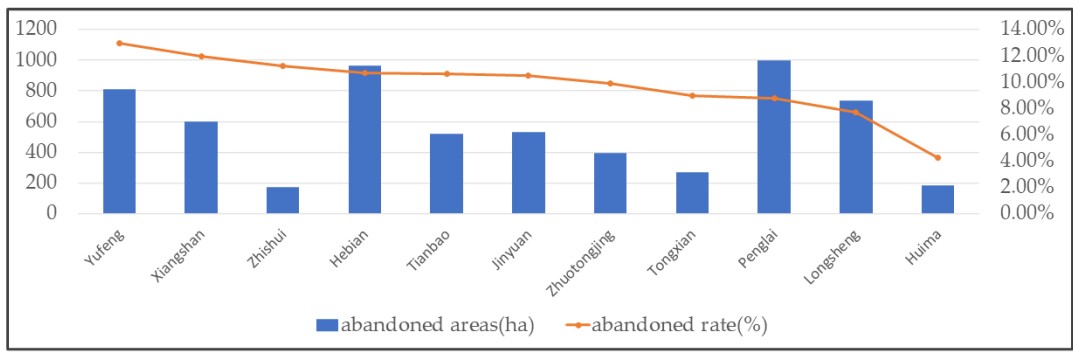

**Figure 9.** The statistics of the abandoned land area and the proportion.

## 4. Discussion

### 4.1. Multi-Source Remote Sensing Image Fusion

Based on multi-source remote sensing data processing, Ls+MVC+FSDAF was used to perform a spatio-temporal fusion for different types of remote sensing images, making full use of the 10 m spatial resolution of Sentinel-2 and the 1-day temporal resolution of MODIS and obtaining the monthly 10 m spatial resolution VIs image set. Moreover, we compared six stretching methods, namely linear stretching, secondary stretching, exponential stretching, geometric stretching, hyperbolic stretching, and logarithmic stretching when fitting Sentinel-2 and MODIS. It was found that for any two images to be stretched, the fitting effect of the linear stretch was always the simplest and most effective. Ls+MVC+FSDAF removed the influence of cloud interference, increased the available remote sensing images, and improved the classification accuracy of the abandoned land. The method may have a good application prospect in surface vegetation coverage and monitoring.

In order to further quantitatively evaluate the accuracy of the MVC, Ls+MVC, and FSDAF, we selected an area of 1000*1000 pixels in the study area and randomly generated images including cloud pixels ratio of 20%, 40%, 60%, and 80%. By comparing the original image and the fused image, the coefficient of determination ($R^2$) and the root mean square error (RMSE) were selected for error evaluation (Table 4). The larger $R^2$ and the smaller RMSE indicated that the fusion effect was more competent. The results show that with the growth of the cloud ratio, $R^2$ decreased and RMSE increased gradually. $R^2$ was 0.8766 and RMSE was 0.0274 after FSDAF, indicating that the results with FSDAF were more credible. The data illustrate that combining the Ls+MVC+FSDAF spatio-temporal fusion algorithm significantly improved the remote sensing spatio-temporal fusion effect. Most studies have shown that the more images of the same area that are obtained, the higher accuracy of the extraction of land cover types [7,39,42,68]. Ls+MVC+FSDAF could fuse multi-source remote sensing images, increase the availability of cloud-contaminated images, and improve the extraction accuracy. However, it is worth noting that some areas were continuously affected by cloud interference during the rainy season for a long time. Neither the Sentinel-2 nor MODIS data were able to eliminate the impact of this continuous cloud coverage. At this time, the use of optical remote sensing images was limited and the fusion of the radar images may be an effective analytical method.

**Table 4.** Comparison of the accuracy of the different methods.

| Fusion Method | Basic Image | S20201112 Cloud Content | $R^2$ | RMSE |
|---|---|---|---|---|
| MVC | S20201107 S20201112 | 20% | 0.9155 | 0.0321 |
| | | 40% | 0.9044 | 0.0326 |
| | | 60% | 0.8988 | 0.0328 |
| | | 80% | 0.8939 | 0.0331 |
| Ls+MVC | S20201112 S20201107 | 20% | 0.9311 | 0.0192 |
| | | 40% | 0.9122 | 0.0214 |
| | | 60% | 0.9020 | 0.0227 |
| | | 80% | 0.8947 | 0.0236 |
| FSDAF | M20201112 S20201107 S20201112 | 20% 40% 60% | 0.8766 | 0.0274 |

### 4.2. Analysis and Suggestions

On the ground of field investigation and visitation, it was found that the reasons for the abandonment of the land in the study area were as follows: (1) land parcels were fragmented, many of which were far away from farmers' houses; (2) some hilly areas had a poor cultivated land quality, large DEM differences, and high slopes, which are not suitable for cultivated land planting. For example, the DEM differences and slopes of Hebian town were higher than that of other towns, and its abandoned area and proportion were high;

(3) weak water conservancy facilities in rural land increased the difficulty of farming and reduced output and efficiency. For example, Huima town is located by the Fujiang river and is rich in water resources. As a result, its abandoned area and proportion were the lowest in the study area. In contrast, the proportion of abandoned land in other towns was significantly higher.

To solve these problems, we propose the following suggestions: (1) cultivated land leasing should be encouraged to increase the turnover rate so that farmers can mainly farm the cultivated land around their houses and focus on contiguous areas to reduce the fragmentation of land; (2) the land must be used rationally from the perspective of ecological environment protection, and scientific and reasonable ecological planning must be implemented. The replanting of abandoned land after logging and fires needs to be strengthened; (3) investment in the construction of water conservancy facilities should be increased in rural areas.

### 4.3. Prospects and Limitations

By integrating Ls, MVC, and FSDAF to fuse multi-source remote sensing images, a collection of remote sensing images with a high spatio-temporal resolution was obtained. They exhibited an excellent performance with high accuracy in the extraction of abandoned land in hilly areas, as well as other land covers, despite the fragmented land and severe cloud interference. In the selection of multi-source remote sensing images, n Sentinel-2 and MODIS images could not only be selected, but also the image pair could be adjusted according to the respective needs of the research to achieve the application effect.

Ls+MVC+FSDAF requires more remote sensing images, which takes a long time for data processing, especially in pixel registration between different images [67]. The accuracy of the remote sensing image registration has an impact on the result of remote sensing image fusion. Adding the automatic registration function of different image pixels can improve the efficiency of image processing and the accuracy of data fusion [68,69]. This study used ENVI software to align the boundaries of different images to achieve the effect of registration. Ls+MVC+FSDAF is based on pixel-level processing; thus, the demand for computing power will rise exponentially with the increase in the research area size and the time sequence. For example, we spent more than 20 h in remote sensing image fusion in the study area in total (The number of pixels was $4325 \times 3350 \times 8 \times 3$. Where $4325 \times 3350$ is the number of pixels in 1 image, 8 is the number of times that need to fuse, and 3 is the number of bands of 1 image. Central processing unit (CPU): AMD R5 3600 @ 3.6 GHz; installed memory (RAM): 16 GB; system type: 64-bit operating system, x64-based processor; operating system: Windows 10). Google Earth Engine (GEE) is a cloud-based platform for planetary-scale geospatial analysis that brings Google's significant computational capabilities to bear on a variety of high-impact societal issues including deforestation, drought, disaster, disease, food security, water management, climate monitoring, and environmental protection [70]. Improving algorithms and using cloud platform processing methods can increase processing efficiency [71,72].

Three vegetation indices based on ndvi, savi, and ndwi are widely used in existing research. However, the situation of abandoned land is much more complicated than that of a single vegetation cover. Further excavating of the variation of band information over time is expected to improve the extraction accuracy of abandoned land. Furthermore, with the development of deep learning which has great potential in the extraction of abandoned land, machine learning methods may be replaced.

The land parcels in the study area were small. Although Sentinel-2 images are public and free with the highest resolution, the mixed pixels still exist, which have a negative impact on the extraction of abandoned land. In the follow-up research, it will be necessary to combine the mixed pixel decomposition model to improve the extraction accuracy of the abandoned land. In this study, the mask was processed based on the existing data, and the accuracy of the mask was also linked to the extraction accuracy of abandoned land. With more accurate base data, high-quality extraction results can be obtained.

## 5. Conclusions

In this study, Sentinel-2, MOD09GA, and MOD13Q1 were used as remote sensing data sources. The ndvi, savi, and ndwi with a monthly spatial resolution of 10 m in the study area were obtained by Ls+MVC+FSDAF. It provided a reference for the rapid extraction of abandoned land in hilly areas that are severely polluted by clouds and planted with diverse vegetation and have small land parcels. Moreover, the abandoned land was extracted based on the technical flowchart of this research, which could provide reliable data support for local food production and land resource management. In addition, based on the spatial distribution of abandoned land and field surveys, reasonable suggestions were put forward to improve local planting conditions and provide technical support for the prosperity of the local economy.

To further extract the spatio-temporal distribution of abandoned land on a larger scale and for a longer time series, big data cloud processing platforms such as GEE will become important tools. Cloud removal algorithms, multi-source remote sensing data fusion algorithms, and deep learning classification models will further improve the accuracy of abandoned land extraction.

**Author Contributions:** Conceptualization, S.H., H.S., W.X., and J.Q.; Formal analysis, S.H., H.S., and W.X.; Funding acquisition, H.S. and W.X.; Investigation, S.H., S.Z., and J.Z.; Methodology, S.H., H.S., and W.X.; Validation, S.H., S.Z., and J.Z.; Writing—original draft, S.H. and W.X.; Writing—review and editing, S.H. and J.Q. All authors have read and agreed to the published version of the manuscript.

**Funding:** This study was supported by the National Natural Science Fund of China (Grant No. 41401659) and the Science and Technology Department of Sichuan Province (Grant No. 2015JY0145).

**Institutional Review Board Statement:** Not applicable.

**Informed Consent Statement:** Not applicable.

**Data Availability Statement:** Sentinel-2 is available via the ESA. MOD09GA and MOD13Q1 are available via USGS. The Openstreetmap vector is available via Openstreetmap official website. Google orthophoto is available via Bigmap platform. The land cover product is available via Earth Science Big Data Science Engineering Data Sharing Service System. The land use distribution map is available via the Resource and Environment Data Cloud Platform.

**Acknowledgments:** The authors are thankful for the provision of the codes of FSDAF and the valuable suggestions by Xiaolin Zhu. This study was supported by the National Natural Science Fund of China (Grant No. 41401659) and the Science and Technology Department of Sichuan Province (Grant No. 2015JY0145).

**Conflicts of Interest:** The authors declare no conflict of interest.

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
