# Peer review of "Extraction of Abandoned Land in Hilly Areas Based on the Spatio-Temporal Fusion of Multi-Source Remote Sensing Images"

_remotesensing, doi:10.3390/rs13193956_

Round 1
Reviewer 1 Report
The article is moderately interesting. Done mainly to determine minimizing the effect of clouds on remote sensing results. My impression is that the identification of areas of land use abandonment was a secondary thing, or at least - it did not advance the main purpose of the research and its description. The description of the research and the algorithms used was presented in a marginal way. The authors hardly give the methodology of their work, except for a few mathematical formulas. Figure 1 gives an overview of the research conducted, but unfortunately does not explain anything to the reader. The figure is not very communicative. I would have seen there the introduction of additional explanations on the lines connecting the different blocks of data. I don't know if this is some sort of process or filter, or simply passing data to subsequent functions or processes of the model.
Because of the great amount of work: collecting a sizeable database of training fields, I think that the article can be proceeded in the journal, but as far as the way of presenting the methodology is concerned - it is laconic and insufficient. In this form it cannot be published.
I am not sure of this comment, but in Fig. 2, cities instead of provinces are listed, e.g., Daying county and nearby Shehong city. A city is usually a point rather than an area as in the figure. Likewise other regions - please check for confusion there. If, on the other hand, the city is so large that it covers the entire area, then a similar section of the orthophoto should be included for a closer representation of the nature of the study area in this publication. However, I have an impression that most of the area is uninhabited and these are not towns but agricultural or cultivated areas or forests.
In Figure 4, it is difficult to tell whether the photographs shown actually represent agriculturally abandoned land. The photographs should be clearer. The authors also did not specify the nature of the samples, what they look like in radar imaging, as in invisible bands and as in UAV images. What is included in Figure 4 is unreadable and does not aid in understanding. How am I supposed to read the data from the Jan --- Dec description when the layers are opaque, overlapping and the first layer covers the others? At first glance I thought it was some form of drawing a 3D slice of the landform. The figure in this form is amendable. To me it makes no sense, because it does not show e.g. abandonment of crops for a longer period, e.g. several years. In this engraving only the scale of the main map is given, no scale of these side maps or scale of ground photographs is given.
Now a few words about combining data. You use the word "fusion" in the publication for something it is not. By this word, you mean something that is fused permanently, e.g. Sentinel-2, MOD09GA and MOD13Q1 fuse their images permanently and offer users the fused images as a product. In your case this is not the case. It is just a combination of some data that is necessary in some situations and not necessary in others. It is not a merger. I ask that you find a different formulation.
Minor comments:
1. please clarify what is meant by line 2 to 3: "Significant marginal phenomenon can 12 be observed in some hilly areas, leading to serious land abandonment." Is this phenomenon significant or marginal?
2. explain the terms spatiotemporal fusion approach of integrating Linear stretch(Ls), maximum value composite (MVC), and Flexible Spatiotemporal DAta Fusion (FSDAF) (lines 104-106) more clearly,
3. the content in lines 442-462 is an outlier from the research conducted. It is completely redundant and unnecessary in this publication. After all, you have not conducted research in this area. It is random content trying to explain in a very general and not very scientific way (more general and common knowledge) the reasons for the occurrence of such a situation in the studied area. If this can be analyzed in social sciences and driving forces, why do you conduct photogrammetric research? In my opinion, the content without research coverage should be removed.
4. the content in lines 272-480 is questionable, namely - you have not conducted research on the efficiency of processing large and changing data, so how can you draw such conclusions. If you want to keep this content - please do some research as to the timing and use of hardware resources for different research areas,
5. please explain the concept of GEE (line 507), with a link, source, etc., please do the same with all data sources used in the publication and research, please further explain the role that Google orthophotos played.
Please mark all changes in the text of the publication with a different font color, and mark deleted content with a strikethrough.
Author Response
Dear Reviewer:
Thank you for your comments concerning our manuscript entitled “Extraction of Abandoned Land in Hilly Areas Based on Spatio-temporal Fusion of Multi-source Remote Sensing Images”(ID:1366741). Those comments are all valuable and very helpful for revising and improving our paper, as well as the important guiding significance to our researches. We have studied comments carefully and have made corrections which we hope meet with approval. Revised portions are marked in red on the paper. The responses to the comments are as flowing:
Point 1: The description of the research and the algorithms used was presented in a marginal way. The authors hardly give the methodology of their work, except for a few mathematical formulas.
Response 1: We added the methodology of MVC in lines 248-252 and the corresponding references [64-65]. At the same time, We added the methodology of Ls and R2 in lines 259-274, and corresponding references [16] and [66].
Point 2: Figure 1 gives an overview of the research conducted, but unfortunately does not explain anything to the reader. The figure is not very communicative.
Response 2: We modified Figure 1 and fully explained each step in lines 116-144.
Point 3:I would have seen there the introduction of additional explanations on the lines connecting the different blocks of data. I don't know if this is some sort of process or filter, or simply passing data to subsequent functions or processes of the model.
Response 3: In the manuscript, there is an inconsistency in the data presented. We have unified the presentation of the data in the introduction and materials in lines 108-109, line 120, and Figure1.
Point 4: Because of the great amount of work: collecting a sizeable database of training fields, I think that the article can be proceeded in the journal, but as far as the way of presenting the methodology is concerned - it is laconic and insufficient. In this form it cannot be published.
Response 4: We modified Figure 4. We have added descriptions of data characteristics and data proceeding methods in lines 178-190 and lines 201-209. In addition, we have performed Jeffries–Matusita distance analysis on the acquired samples in lines 203-206 and added corresponding references [62].
Point 5: I am not sure of this comment, but in Fig. 2, cities instead of provinces are listed, e.g., Daying county and nearby Shehong city. A city is usually a point rather than an area as in the figure. Likewise other regions - please check for confusion there. If, on the other hand, the city is so large that it covers the entire area, then a similar section of the orthophoto should be included for a closer representation of the nature of the study area in this publication. However, I have an impression that most of the area is uninhabited and these are not towns but agricultural or cultivated areas or forests.
Response 5: We have modified Figure 2 to avoid confusion when reading. Furthermore, we added the DEM map of the study area in Figure 2 to analyze the relationship between the abandoned land and DEM in the later result.
Point 6: In Figure 4, it is difficult to tell whether the photographs shown actually represent agriculturally abandoned land. The photographs should be clearer. The authors also did not specify the nature of the samples, what they look like in radar imaging, as in invisible bands and as in UAV images. What is included in Figure 4 is unreadable and does not aid in understanding. How am I supposed to read the data from the Jan --- Dec description when the layers are opaque, overlapping and the first layer covers the others? At first glance I thought it was some form of drawing a 3D slice of the landform. The figure in this form is amendable. To me it makes no sense, because it does not show e.g. abandonment of crops for a longer period, e.g. several years. In this engraving only the scale of the main map is given, no scale of these side maps or scale of ground photographs is given.
Response 6: We have modified Figure 4. The scale of the side maps has been added, and the description of the appearance of the samples on different images has also been added in lines 178-190.
Point 7: Now a few words about combining data. You use the word "fusion" in the publication for something it is not. By this word, you mean something that is fused permanently, e.g. Sentinel-2, MOD09GA and MOD13Q1 fuse their images permanently and offer users the fused images as a product. In your case this is not the case. It is just a combination of some data that is necessary in some situations and not necessary in others. It is not a merger. I ask that you find a different formulation.
Response 7: We modify ‘Ls-m+FSDAF’ to ‘Ls+MVC+FSDAF’ in the full text, and modify some 'fuse' in the text to 'combine' or 'integrate'.
Point 8: please clarify what is meant by line 2 to 3: "Significant marginal phenomenon can be observed in some hilly areas, leading to serious land abandonment." Is this phenomenon significant or marginal?
Response 8: This is a marginal phenomenon. We have deleted the word of 'Significant'.
Point 9: explain the terms spatiotemporal fusion approach of integrating Linear stretch(Ls), maximum value composite (MVC), and Flexible Spatiotemporal DAta Fusion (FSDAF) (lines 104-106) more clearly.
Response 9: We added the methodology of MVC in lines 248-252 and the corresponding references [64-65]. At the same time, We added the methodology of Ls and R2 in lines 259-274, and corresponding references [16] and [66].
Point 10: the content in lines 442-462 is an outlier from the research conducted. It is completely redundant and unnecessary in this publication. After all, you have not conducted research in this area. It is random content trying to explain in a very general and not very scientific way (more general and common knowledge) the reasons for the occurrence of such a situation in the studied area. If this can be analyzed in social sciences and driving forces, why do you conduct photogrammetric research? In my opinion, the content without research coverage should be removed.
Response 10: We deleted the part of the social sciences and driving forces. Based on the characteristics of the study area, reasonable suggestions were made from the perspectives of land fragmentation, water resources distribution, and DEM in lines 486-504.
Point 11: the content in lines 272-480 is questionable, namely - you have not conducted research on the efficiency of processing large and changing data, so how can you draw such conclusions. If you want to keep this content - please do some research as to the timing and use of hardware resources for different research areas.
Response 11: We have added the part of conducted research on the efficiency in lines 522-527. In addition, reference [69-70] shows that spatial-temporal fusion requires a lot of computing power, and reference [71-73] shows that GEE has strong computing power and has been widely used in various fields of remote sensing monitoring.
Point 12: please explain the concept of GEE (line 507), with a link, source, etc., please do the same with all data sources used in the publication and research, please further explain the role that Google orthophotos played.
Response 12: We have added the concept of GEE in lines 527-532 and reference [71]. We have added all data sources used in the publication and research in lines 164-167 and reference [58-61]. We have further explained the role that Google orthophotos played in lines 123-135 and lines 181-190.

Reviewer 2 Report
In this paper, a new spatiotemporal fusion approach of integrating Linear stretch (Ls), maximum value composite (MVC), and Flexible Spatiotemporal DAta Fusion (FSDAF) were proposed to analyze the time-series changes and extract the spatial distribution of abandoned land. The paper clearly described the approach, carefully applies the methodology, and comes up with sound results, which are well documented. I recommend the paper to be published with very minor revisions. Below are my minor suggestions for improving the paper.
- Rearrange the keywords alphabetically.
- Why both the land use distribution map in 2018 and land cover product in 2020 was simultaneously
- The points color of woodland in Fig4 should be change to other for better identification.
- L192 explicit projection category.
- The final resolution of results are 10m?
- L193 which resample method was adopts and write it in paper
- Resample MOD09GA and MOD13Q1 images in the near-infrared band and red band to a spatial resolution of 10 meters, and the mid-infrared band to 20 meters; perform Ls-m+FSDAF fusion on ndwi to obtain a 20m spatial resolution image and resampled to 10m spatial resolution. Why mid-infrared band directly resample to 10m?
- How many landcover in study area, just three types (100 woodland samples, 100 cultivated land samples, and 100 abandoned land samples) are selected is enough?
- Each abandoned land areas in Figure8 should be given with the bar graph.
- Please, see further comments in the attached file.

Author Response
Dear Reviewer:
Thank you for your comments concerning our manuscript entitled “Extraction of Abandoned Land in Hilly Areas Based on Spatio-temporal Fusion of Multi-source Remote Sensing Images”(ID:1366741). Those comments are all valuable and very helpful for revising and improving our paper, as well as the important guiding significance to our researches. We have studied comments carefully and have made corrections which we hope meet with approval. Revised portions are marked in red on the paper. The responses to the comments are as flowing:
Point 1: Rearrange the keywords alphabetically.
Response 1: We rearranged the keywords in alphabetical order.
Point 2: Why both the land use distribution map in 2018 and land cover product in 2020 was simultaneously.
Response 2: The two data are from different departments. The land use distribution map in 2018 comes from the Sichuan Provincial Department of Natural Resources. The land cover product in 2020 is a global land classification product. The overlay analysis of the two is used for Improve the accuracy of woodland and cultivated land.
Point 3: The points color of woodland in Fig4 should be change to other for better identification.
Response 3: We modified the color of woodland by the color scheme of land use products.
Point 4: L192 explicit projection category.
Response 4: We have increased the type of projection.
Point 5: The final resolution of results are 10m.
Response 5: Yes, we got a final resolution of 10 meters.
Point 6: L193 which resample method was adopts and write it in paper.
Response 6: We have added a method of resampling, that is, the bilinear method.
Point 7: Resample MOD09GA and MOD13Q1 images in the near-infrared band and red band to a spatial resolution of 10 meters, and the mid-infrared band to 20 meters; perform Ls-m+FSDAF fusion on ndwi to obtain a 20m spatial resolution image and resampled to 10m spatial resolution. Why mid-infrared band directly resample to 10m?
Response 7: Since MOD09GA and MOD13Q1 images in mid-infrared band is 500m (Table 2), Sentinel-2 images in mid-infraed band is 20m. Their spatial resolution ratio is 25. If we resample directly to 10m, the spatial resolution ratio Increase. When the spatial resolution ratio is greater than 30, the fusion effect is not ideal(reference [23]). So, we fuse images first and then resampled them. After resampling, the pixel size of ndwi is consistent with ndvi and savi.
Point 8: How many landcover in study area, just three types (100 woodland samples, 100 cultivated land samples, and 100 abandoned land samples) are selected is enough?
Response 8: As shown in Figure 1, we have extracted the range of woodland and cultivated land firstly. The target of our research is the abandoned land in the woodland and cultivated land. Therefore, according to the research goal, we selected the sample data of these three types of land use.
Point 9: Each abandoned land areas in Figure8 should be given with the bar graph.
Response 9: The area of each abandoned land in Figure 8 has been given as a bar graph in Figure 9.
Point 10: Please, see further comments in the attached file.
Response 10: We made further modifications to each comment.

Reviewer 3 Report
Title: Extraction of Abandoned Land in Hilly Areas Based on Spatio-temporal Fusion of Multi-source Remote Sensing Images
- The instruction for authors: The abstract should be written as one paragraph (of about 300 words). The abstract must state the overarching goal or main objective(s) as well as the purpose/rationale of the study in two or three sentences. This should be followed by succinct statements concerning the data used, approach adopted, and methods used. The manuscript abstract is too big and should be corrected like recommendation.
- The key word “Hilly land” should be changed to “Hilly area” because this term is using in the manuscript. It is also not recommended to choose keywords that are in the title.
- Line 152, 156: The web sites of data set should change to citation like “[10, 11].” and add to the reference list ”Scihub Copennicus,…. 2021”. Data MOD09GA and MOD13Q1 are small resolution. The authors write “abandoned land in a small area is usually mapped, using satellite images with high and/or medium resolution...”(line 59). Why you used it in the study?
- The authors should input the purpose of the article in the end of introduction.
- The authors used large enough the references list. The article is interesting and after minor correction could be publish in the special issue “Remote Sensing Modeling and Retrieving for Mountain Ecological Parameters”.
Author Response
Dear Reviewer:
Thank you for your comments concerning our manuscript entitled “Extraction of Abandoned Land in Hilly Areas Based on Spatio-temporal Fusion of Multi-source Remote Sensing Images”(ID:1366741). Those comments are all valuable and very helpful for revising and improving our paper, as well as the important guiding significance to our researches. We have studied comments carefully and have made corrections which we hope meet with approval. Revised portions are marked in red on the paper. The responses to the comments are as flowing:
Point 1: The instruction for authors: The abstract should be written as one paragraph (of about 300 words). The abstract must state the overarching goal or main objective(s) as well as the purpose/rationale of the study in two or three sentences. This should be followed by succinct statements concerning the data used, approach adopted, and methods used. The manuscript abstract is too big and should be corrected like recommendation.
Response 1: We further revised the abstract part as required.
Point 2: The key word “Hilly land” should be changed to “Hilly area” because this term is using in the manuscript. It is also not recommended to choose keywords that are in the title.
Response 2: We modified ‘Hilly land’ to ‘Hilly area’.
Point 3: Line 152, 156: The web sites of data set should change to citation like “[10, 11].” and add to the reference list ”Scihub Copennicus,…. 2021.
Response 3: We have revised the statement in this part and added the corresponding data reference source.
Point 4: Data MOD09GA and MOD13Q1 are small resolution. The authors write “abandoned land in a small area is usually mapped, using satellite images with high and/or medium resolution...”(line 59). Why you used it in the study?
Response 4: Because MODIS data has a high temporal resolution, it is widely used in spatio-temporal fusion, reference[8-21].
Point 5: The authors should input the purpose of the article in the end of introduction.
Response 5: We added the research purpose at the end of the introduction(lines 111-113).
Point 6: The authors used large enough the references list. The article is interesting and after minor correction could be publish in the special issue “Remote Sensing Modeling and Retrieving for Mountain Ecological Parameters.
Response 6: Thank you very much for your recognition, it is a great motivation for us.

Round 2
Reviewer 1 Report
Thank you for your positive response to my review. Currently, in my opinion, the article can be published. Thank you and congratulations.